# STEP-UNROLLED DENOISING AUTOENCODERS FOR TEXT GENERATION

**Nikolay Savinov**[*]    **Junyoung Chung**[*]    **Mikołaj Bińkowski**[*]
**Erich Elsen**    **Aäron van den Oord**
DeepMind, London, UK

## ABSTRACT

In this paper we propose a new generative model of text, *Step-unrolled Denoising Autoencoder* (SUNDAE), that does not rely on autoregressive models. Similarly to denoising diffusion techniques, SUNDAE is repeatedly applied on a sequence of tokens, starting from random inputs and improving them each time until convergence. We present a simple new improvement operator that converges in fewer iterations than diffusion methods, while qualitatively producing better samples on natural language datasets. SUNDAE achieves state-of-the-art results (among non-autoregressive methods) on the WMT'14 English-to-German translation task and good qualitative results on unconditional language modeling on the Colossal Cleaned Common Crawl dataset and a dataset of Python code from GitHub. The non-autoregressive nature of SUNDAE opens up possibilities beyond left-to-right prompted generation, by filling in arbitrary blank patterns in a template.

## 1 INTRODUCTION

Autoregressive (AR) models have shown excellent results in generating text (e.g., GPT-3, Brown et al., 2020). However, while their training scales very well, sampling is prohibitively slow for many practical applications. Moreover, there are limitations to the kinds of conditioning AR models can seamlessly handle: the left-to-right restriction makes it hard to "fill in the gaps" in a partially written text draft. Even more importantly, this prohibits iterative refinement of complete text drafts to make them more self-consistent, which is a common task for human writers. Finally, AR models require network architectures to be causal, severely limiting the kinds of neural network architectures that can be used for text-modeling. All of these motivated the machine learning community to make extensive efforts to propose alternatives to AR models.

Machine translation (MT) was perhaps one of the first tasks where non-AR approaches were shown to seriously rival the AR-based state of the art: methods like CMLM (Ghazvininejad et al., 2019) and DisCo (Kasai et al., 2020) show promising results and their decoding speed is excellent compared to AR. However, while their performance is competitive, they are still behind the AR benchmark and actually require distillation of a larger AR model — without which, performance drops considerably.

Non-AR methods have proven hard to apply to the general unconditional language modeling (LM) task. When there is no conditioning, the multi-modality problem becomes paramount, as shown by Gu et al. (2017), which likely makes it problematic to use methods like CMLM and DisCo because their decoding mechanism is deterministic and does not model uncertainty. Yet, recently the community has seen promising results from non-AR models like Multinomial Diffusion (Hoogeboom et al., 2021) and D3PM (Austin et al., 2021). These methods optimize a lower bound (ELBO) on likelihoods and have shown negative log-likelihood (NLL) results approaching AR models on several benchmarks like text8 (Mahoney, 2011) and LM1B (Chelba et al., 2013). However, a major gap in NLL persists, and samples from those models lack coherence.

In this paper we propose a novel non-autoregressive method which shows state-of-the-art results in machine translation on WMT'14 EN→DE raw data (without distillation from AR) amongst non-AR methods and good qualitative results on unconditional language modeling on the Colossal Clean Common Crawl (C4) dataset (Raffel et al., 2019) and a dataset of Python code from GitHub. Our model operates as a time-homogeneous Markov Chain similar to that of Lee et al. (2018): conditioned on the corrupted data, it tries to approximate the original uncorrupted samples by a per-token

---

[*]Shared first authorship.

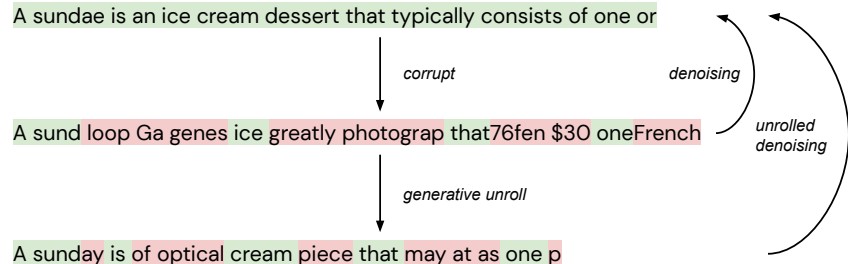

Figure 1: The difference between denoising and *unrolled denoising*. Original text (top) is randomly corrupted, producing a text (middle) where some tokens are original (green) and others are corrupted (red). This text is denoised by sampling from the generative model to produce another noisy text (bottom). While standard denoising autoencoders only learn a mapping from the middle text to the top text, *Step-unrolled Denoising Autoencoder* learns a mapping from bottom to top (including middle as a special case of zero unroll steps). This has an intuitive meaning: during generation time, the network will mostly (right after the first step) encounter texts like the bottom one, not like the middle one — so unrolls prepare the model during training for inputs it will get at generation time.

factorised distribution. During generation, we unroll the chain by sampling from the transition distribution and feeding samples back into the input. We propose a new training mechanism called *unrolled denoising*, which uses unrolls of the chain during training as well, and which we empirically show to be crucial for practical performance of the algorithm in ablation studies. This feature motivates the name for the model, *Step-unrolled Denoising Autoencoder* (SUNDAE). The difference between usual denoising and *unrolled denoising* is illustrated in Figure 1.

To summarize our contributions:

- We present SUNDAE, a new generative model of text that unrolls the denoising process during training.
- SUNDAE achieves state-of-the-art results on WMT'14 English-to-German translation task among non-AR methods.
- We demonstrate good qualitative results for unconditional generation and inpainting on Colossal Clean Common Crawl dataset and a dataset of Python code from GitHub.
- We carefully ablate and analyze the properties of the proposed method, and show that unrolls during training are crucial for the model's performance.

## 2  METHOD

We approach the problem of generative modelling of discrete sequences by bringing together the framework of denoising autoencoders and Markov chain models. In this section, we first discuss the definition of the generative model, and then describe the training process, which includes our main contribution, *unrolled denoising*.

For a fixed prior distribution $p_0$ on some space $X$ consider a process $\mathbf{x}_t \sim f_\theta(\cdot|\mathbf{x}_{t-1})$, where $f_\theta$ is a parametric transition function. Then $\{\mathbf{x}_t\}_t$ is a time-homogeneous Markov chain, and transition $t$-steps ahead has the following form[1]

$$p_t(\boldsymbol{x}_t|\boldsymbol{x}_0) = \sum_{\boldsymbol{x}_1, \boldsymbol{x}_2, \ldots, \boldsymbol{x}_{t-1} \in X} \prod_{s=1}^{t} f_\theta(\boldsymbol{x}_s|\boldsymbol{x}_{s-1}). \qquad (1)$$

For a fixed number of steps $T$, the prior $p_0$ and decoder $p_T$ together determine our model distribution $p_T(\mathbf{x}_T) = p_T(\mathbf{x}_T|\mathbf{x}_0)p_0(\mathbf{x}_0)$.

We assume $X = \{1, \ldots, v\}^N$ to be the space of sequences of length $N$ with values at all positions coming from a vocabulary of size $v$, and the prior $p_0$ to be uniform over $X$. Let $p_{\text{data}}$ be a distribution of the data and assume that $f_\theta$ induces a conditional probability distribution $f_\theta(\cdot|\boldsymbol{x}')$ that can be

---

[1]for formal derivation, see Appendix A.

factorised into a conditionally-indepedent product as follows:

$$f_\theta(\boldsymbol{x}|\boldsymbol{x}') = f_\theta^{(1)}(\boldsymbol{x}^{(1)}|\boldsymbol{x}')f_\theta^{(2)}(\boldsymbol{x}^{(2)}|\boldsymbol{x}')\cdots f_\theta^{(N)}(\boldsymbol{x}^{(N)}|\boldsymbol{x}'), \qquad \text{for } \boldsymbol{x},\boldsymbol{x}' \in X. \tag{2}$$

Note that although for $t = 1$ the distribution $p_1(\cdot|\boldsymbol{x}_0)$ is limited by product structure (Eq. 2), the subsequent $p_t$s are not restricted in the same way, and after each step they belong to potentially more and more expressive family.

## 2.1 TRAINING WITH UNROLLED DENOISING

Due to intractability of the likelihood of the model $p_T$ as well as the computational requirements of optimising the whole chain $\{p_t\}_t$, we propose an efficient two-step training method which we term *unrolled denosing* and visualise in Figure 1. We consider a smaller number of steps than $T$ that we intend to use at sampling, but to compensate for that, we unroll the chain starting from corrupted data samples, rather than the prior $p_0$. This way, the model learns to denoise the samples it is likely to encounter during the full unroll used at sample time. While using a single step would resemble the training strategy of BERT (Devlin et al., 2018), using at least two steps is essential for the performance of our model.

Consider a *corruption* distribution $q(\boldsymbol{x}^c|\mathbf{x})$ that replaces a random proportion of tokens in a sentence $\mathbf{x}$ with ones randomly sampled from a uniform distribution. Our objective is given by $L^{(1:2)} = \frac{1}{2}\left(L^{(1)} + L^{(2)}\right)$ where

$$L^{(t)}(\theta) := -\mathbb{E}_{\substack{\mathbf{x}\sim p_{\text{data}} \\ \mathbf{x}_0\sim q(\cdot|\mathbf{x}) \\ \mathbf{x}_1\sim f_\theta(\cdot|\mathbf{x}_0) \\ \cdots \\ \mathbf{x}_{t-1}\sim f_\theta(\cdot|\mathbf{x}_{t-2})}} [\log f_\theta(\mathbf{x}|\mathbf{x}_{t-1})], \tag{3}$$

i.e. the reconstruction loss of the chain after $t$ steps, starting from a corrupted sample $\mathbf{x}_0$. In Appendix A.1 we show that $L^{(t)}$ is an upper bound on the actual negative log-likelihood from the model distribution $p_t$:

$$\widetilde{L^{(t)}}(\theta) := -\mathbb{E}_{\substack{\mathbf{x}\sim p_{\text{data}} \\ \mathbf{x}^c\sim q(\cdot|\mathbf{x})}} [\log p_t(\mathbf{x}|\mathbf{x}^c)]. \tag{4}$$

Note that $\widetilde{L^{(T)}}$ is the actual term we would like to minimise if $p_T$ were tractable, and in fact is closely related to evidence lower bound (ELBO) via a fixed encoder $q(\mathbf{x}^c|\mathbf{x})$ (Appendix A.2). However, due to the inherent non-differentiable nature of discrete Markov chain models, optimisation of $\widetilde{L^{(T)}}$ would not only be costly, but also limit the gradient flow to the last step of the chain. We found instead that optimising several steps together without propagating the gradient through sampling is sufficient for obtaining good results. Averaging more $L^{(t)}$ terms can lead to minor improvements in performance (see Figure 4d in Appendix C), but it considerably slows down the training speed. We discuss these and other loss variants further in Section 3.1.

Since function $f_\theta$ is modelled as a neural network, the logarithm under expectation on the right hand side of Eq. 3 is obtained from predicted logits, thus we term $L^{(1)}(\theta)$ and $L^{(2)}(\theta)$ *logits* loss and *unrolled logits* loss, respectively.

**Corruption function.** To corrupt texts, we first sample a proportion of tokens uniformly from $[0, 1]$, randomly select positions according to this proportion and then change tokens at those positions to random tokens sampled uniformly from the vocabulary. This way, we want to ensure that samples seen at various steps during a generative unroll (including the first one, sampled from the uniform prior) are well-represented in corruptions applied during training. More details and relations to the forward diffusion process (Hoogeboom et al., 2021) are presented in Appendix A.3.

## 2.2 SAMPLING

At sampling time we follow the structure of the Markov process and sample sequentially $\boldsymbol{x}_t \sim f_\theta(\boldsymbol{x}_t|\boldsymbol{x}_{t-1})$ for some fixed number of steps $T$, beginning from a random sequence $\boldsymbol{x}_0$ (potentially starting with a prompt or a template for conditional generation). To control the speed of convergence, we propose three improved strategies which allow the use of a much smaller number of steps:

- **Low-temperature sampling**. For temperature $\tau$ and trained network $f_\theta$ we consider modified function $f_\theta^\tau$ such that $\log f_\theta^\tau(\cdot|\boldsymbol{x}) \propto \frac{1}{\tau}\log f_\theta(\cdot|\boldsymbol{x})$. We found that sampling with temperatures lower than 1 leads to much faster convergence than with the original logits, and allowed generation of high-quality samples in a matter of 10-16 steps.

| Model | Steps ($T$) | Raw BLEU | | AR-distilled BLEU | |
|---|---|---|---|---|---|
| | | EN→DE | DE→EN | EN→DE | DE→EN |
| **AR Models** | | | | | |
| Transformer Base (65M) (Vaswani et al., 2017) ($n\!=\!4$) | - | 27.3 | 31.78* | - | - |
| **Non-AR Models** | | | | | |
| NAT (Gu et al., 2017) ($n\!=\!100$) | 1 | - | - | 19.17 | 23.20 |
| LVM-DAE (Lee et al., 2018) | - | - | - | 21.54 | 25.43 |
| NAT-REG (Wang et al., 2019) ($n\!=\!9$) | 1 | - | - | 24.61 | 28.90 |
| LV-NAR (Shu et al., 2020) ($n\!=\!50$) | 1 | 11.8 | - | 25.10 | - |
| NART w/ hints (Li et al., 2019)($n\!=\!9$) | 1 | - | - | 25.20 | 29.52 |
| FlowSeq (Ma et al., 2019) ($n\!=\!30$) | 1 | 23.64 | 28.29 | 25.31 | 30.68 |
| ReorderNAT (Ran et al., 2019) | 1 | - | - | 26.49 | 31.13 |
| NART (Sun et al., 2019) ($n\!=\!19$) | 1 | - | - | 26.80 | 30.04 |
| CMLM (Ghazvininejad et al., 2019) + Mask-Predict ($n\!=\!5$) | 4 | 22.25 | - | 25.94 | 29.90 |
| CMLM (Ghazvininejad et al., 2019) + Mask-Predict ($n\!=\!5$) | 10 | 24.61 | - | 27.03 | 30.53 |
| DisCo (Kasai et al., 2020) + Mask-Predict ($n\!=\!5$) | 4 | - | - | 25.83 | 30.15 |
| DisCo (Kasai et al., 2020) + Mask-Predict ($n\!=\!5$) | 10 | - | - | 27.06 | 30.89 |
| DisCo (Kasai et al., 2020) + Easy-First ($n\!=\!5$) | 4-5† | 24.8 | - | 27.34 | 31.31 |
| NARLVM (Lee et al., 2020) ($n\!=\!25$) | 4 | - | - | 27.40 | - |
| JM-NAT (Guo et al., 2020) ($n\!=\!3$) | 4 | - | - | 27.05 | 31.51 |
| JM-NAT (Guo et al., 2020) ($n\!=\!3$) | 10 | - | - | 27.69 | **32.54** |
| SMART (Ghazvininejad et al., 2020) ($n\!=\!5$) | 4 | - | - | 27.03 | 30.87 |
| SMART (Ghazvininejad et al., 2020) ($n\!=\!5$) | 10 | - | - | 27.65 | 31.27 |
| Imputer (Saharia et al., 2020) ($n\!=\!1$) | 4 | 24.7 | - | 28.0 | 31.0 |
| Imputer (Saharia et al., 2020) ($n\!=\!1$) | 8 | 25.2 | - | 28.2 | 31.3 |
| **SUNDAE (ours 63M)** | | | | | |
| Deterministic ($n\!=\!16$) | 4 | 25.01 | 29.53 | 28.33 | 32.25 |
| Deterministic ($n\!=\!16$) | 8 | 25.53 | 30.01 | 28.32 | 32.27 |
| Deterministic ($n\!=\!16$) | 10 | 25.54 | 30.11 | 28.32 | 32.27 |
| Stochastic ($n\!=\!16$) | 4 | 23.05 | 28.13 | 27.94 | 32.10 |
| Stochastic ($n\!=\!16$) | 8 | 26.08 | 30.48 | 28.23 | 32.33 |
| Stochastic ($n\!=\!16$) | 10 | **26.25** | **30.80** | 28.33 | 32.29 |
| Stochastic ($n\!=\!16$) | 16 | 26.24 | 30.76 | **28.46** | 32.30 |

Table 1: Test BLEU scores of AR and non-AR systems on the WMT'14 English-to-German (EN→DE) and German-to-English (DE→EN) translation tasks. The number of reranked candidates is denoted $n$. SUNDAE does not use an AR model for re-ranking. We highlight BLEU scores of the best non-AR systems in bold font. All the entries are ordered based on the EN→DE BLEU score. *indicates the results of the Transformer baselines implemented by the authors of DisCo. †DisCo has a dynamic inference mode for which we report the average number of steps.

- **Argmax-unrolled decoding**. At the limit $\tau \rightarrow 0$ sampling with temperature reduces to deterministic argmax decoding where the most probable token is chosen at each step. Relatedly to our *unrolled logits* loss, we modify this strategy by resampling the low-certainty tokens in accordance with *unrolled logits*. This heuristic allowed further improvements to sampling speed while maintaining the high quality of the samples. We discuss it in detail in Section 3.1.

- **Updating fewer tokens**. In tasks where diversity is paramount (like unconditional text generation), we found that updating a random subset of tokens at each decoding step leads to faster convergence. This likely happens because independent sampling of all tokens might create uncoordinated changes which could take some time to fix in the subsequent steps. We use this strategy in Section 3.2.

# 3 EXPERIMENTS

## 3.1 MACHINE TRANSLATION

We first evaluate SUNDAE on Machine Translation (MT) benchmarks. We compare SUNDAE against AR and non-AR models in terms of the translation quality using BLEU (Papineni et al., 2002) as the metric. We demonstrate that without using techniques like sequence-level knowledge distillation (Kim & Rush, 2016), SUNDAE performs almost as well as the AR model and outperforms all other methods that do not rely on AR models.

**Target Length Prediction.** Unlike their AR counterparts, non-AR models do not explicitly learn to predict an end of sequence, but instead have been shown to benefit from auxiliary target length

prediction (Lee et al., 2018; Ghazvininejad et al., 2019; Kasai et al., 2020). A common approach is to treat it as a classification task, predicting either the exact length or the difference between the source and target lengths. The decoding process has to commit to the predicted length, and often multiple length beams are used to get the best translation results. In our models, we use a separate network that receives source encodings as input and predicts corresponding target length. Target length embedding vector is prepended to source embeddings so that decoder can attend to it. During training, we use the reference target length, while at sampling, a predicted one, although the model does not need to fully commit to the predicted length like some previous non-AR models (Saharia et al., 2020). Note that the length classification loss does not affect the encoder parameters. Our models are also trained to predict the padding tokens at the end of the text. For more details, see Appendix B.

**Experiment Settings.** We conduct experiments on WMT'14 parallel corpora using EN↔DE (4.5M pairs) and EN→FR (36M pairs) translation tasks. The raw texts are encoded using BPE (Sennrich et al., 2015) as the subword units, and we use the same preprocessed data as in Vaswani et al. (2017) for fair comparisons. We evaluate the performance by measuring BLEU (Papineni et al., 2002; Post, 2018) on the test split of each translation task[2]. We use the encoder-decoder Transformer architecture for MT (Vaswani et al., 2017), but remove the causality masking in the decoder. There are 6 attention layers for both encoder and decoder, 8 attention heads, 512 model dimension and 2048 feedforward hidden dimension. The total number of parameters is 63M including the target length prediction module described in 3.1. We use dropout ($p = 0.1$) and label smoothing ($\epsilon = 0.1$) during training for all tasks, except for AR-distilled EN→DE, where we found lower dropout ($p = 0.05$) and no label smoothing produces better results on validation. The training batch size is 4096, and we use Adam (Kingma & Ba, 2014) with $\beta_1 = 0.9$, $\beta_2 = 0.999$, $\epsilon = 10^{-6}$ and weight decay of 0.1 (Loshchilov & Hutter, 2017). We warm up the learning rate from $10^{-7}$ to $10^{-4}$ in the first 5K steps and decay it to $10^{-5}$ using cosine annealing (Loshchilov & Hutter, 2016). Our models were trained for $10^6$ steps using 16 TPU accelerators using bfloat16 precision. We crop sequences that have more than 128 tokens (this occurs less than 0.2% in training data). Finally, we average the last 10 checkpoints to obtain a single model for the evaluation. All hyperparameter tuning is performed on a held-out validation set.

**Decoding.** We decode translations from SUNDAE using two different types of heuristic decoding methods in MT experiments. The decoding process always begins with an array of random integers sampled from the discrete uniform prior $\boldsymbol{y}_0 \sim p_0$, while the encoder input - the source sentence - remains the same throughout the process. The first decoding method is *low-temperature sampling*, where we iteratively sample $\boldsymbol{y}_t \sim f_\theta^\tau(\cdot|\boldsymbol{y}_{t-1})$ for $T$ iterations (with the decoder output logits divided by $\tau$, see Section 2.2), with $T \leq 16$ and $\tau \in [0.1, 0.6]$ determined based on the validation performance. The second method is *argmax-unrolled decoding*, which requires a smaller number of iterations compared to its stochastic counterpart. In argmax-unrolled decoding, we first compute the logits $\lambda_1 = f_\theta(\cdot|\boldsymbol{y}_0)$ of the initial sample array $\boldsymbol{y}_0$ and obtain the samples $\boldsymbol{y}_1$, pass a tuple $(\boldsymbol{y}_1, \lambda_1)$ to the next iteration. At each step $t \geq 2$, the method finds top-$\rho$ share of the tokens that are sorted by the log-probability in descending order (i.e. uncertain tokens) from $\lambda_{t-1}$, where $\rho \in [0.1, 0.6]$ is another hyperparameter searched for in the validation stage. We compute *unrolled logits* for those top-$\rho$ tokens and then apply $\arg\max$ to obtain *unrolled tokens*. For the rest of the input tokens of $\boldsymbol{y}_{t-1}$ we compute logits once and obtain $\lambda_t = f_\theta(\cdot|\boldsymbol{y}_{t-1})$ and then apply $\arg\max$ to obtain *predicted tokens*. Finally, we combine *unrolled tokens* for uncertain input positions with *predicted tokens* for remaining positions to obtain $\boldsymbol{y}_t$, and the tuple $(\boldsymbol{y}_t, \lambda_t)$ is provided as the input to the next iteration. This procedure is repeated over a fixed number of iterations $T$. We always decode $n$ samples in parallel and rerank them based on the model score. We show the relative speed gain of SUNDAE in Table 2, the Transformer base is used as the AR baseline, for which we perform incremental sampling by caching the previous attention states.

**Baselines.** We compare SUNDAE with Transformer base (Vaswani et al., 2017) as the AR baseline and several non-AR models including CMLM (Ghazvininejad et al., 2019) and DisCo (Kasai et al., 2020). The last two are both iterative non-AR models that share a substantial amount of common ground with SUNDAE. However, their best results come from the variants distilled from Transformer large (Vaswani et al., 2017), with only few test results available without distillation. As we aim to remove any kinds of dependencies on AR approaches, we would like to make a distinction

---

[2]Due to limited space, we report BLEU using SacreBLEU in Appendix G.

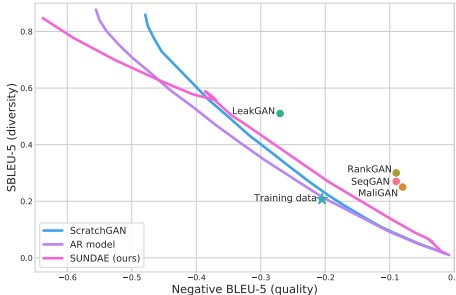

| Steps ($T$) | Relative Speed Improvement |
|:---:|:---:|
| 4 | 4.7x |
| 8 | 2.6x |
| 10 | 2.2x |
| 16 | 1.4x |

Figure 2: BLEU scores on EMNLP2017 News. Left is better, lower is better. Quality/variation is controlled by changing the temperature.

Table 2: Relative speed gain of SUNDAE over AR Transformer base (greedy decoding) on WMT'14 EN→DE validation set.

between raw, fully-non-AR models, and *AR-distilled* ones, i.e. those trained via distillation from large AR teachers (which typically perform better than the student models), or via AR reranking.

**Results and Ablation Studies.** In Table 1, we show test BLEU scores from SUNDAE and other baselines. On EN→DE, SUNDAE achieved 26.25 BLEU, and it performed the best among the raw non-AR models. To the best of our knowledge, SUNDAE achieved the closest result ($\triangle = 1.05$) to the score of Transformer base (Vaswani et al., 2017) without the aid of AR models.[3] In the raw setting, argmax-unrolled decoding outperforms mask-predict (Ghazvininejad et al., 2019) and easy-first (Kasai et al., 2020) by 0.74 BLEU when using the same amount of the worst-case compute budget. At $T = 8$, SUNDAE performs better than Imputer (Saharia et al., 2020), which is a strong baseline that can generate high-quality translation within few (4-8) iterations. We discovered that as $T$ increases, low-temperature sampling outperforms argmax-unrolled decoding, and at $T = 10$, low-temperature sampling outperforms *Mask-Predict* (Ghazvininejad et al., 2019) and *Easy-First* (Kasai et al., 2020) by 1.5 BLEU score. SUNDAE achieved 30.80 BLEU on DE→EN, where there are not that many raw baselines to compare with. The difference between SUNDAE and the AR baseline on DE→EN is 0.98. Finally, SUNDAE achieved 37.53 BLEU on EN→FR at $T = 10$, while the Transformer base (Vaswani et al., 2017) reported 38.1 BLEU on this task. Thus, SUNDAE without distillation is only 0.57 BLEU behind the standard AR baseline on EN→FR. We present more details on EN→FR scores in Table 7 in Appendix G.

While knowledge distillation was outside our main focus, we also report scores for AR-distilled SUNDAE.[4] It again performed well, establishing new SotA on EN→DE translation among AR–distilled models, and being second only to JM-NAT (Guo et al., 2020) in DE→EN task.

We ablate the number of steps of *unrolled denoising* on EN→DE test split. We observe that having at least one *unrolled denoising* step is crucial to obtain good performance in translation. With only the $L^{(1)}$ loss, SUNDAE achieves 11.19 BLEU, whereas using $L^{(1:2)}$, the score improves to 26.57 BLEU. Using $L^{(1:3)}$, i.e. averaging more *unrolled denoising* loss terms, did not improve the performance further, scoring 26.25 BLEU. The target length prediction also turned out to be an important component of our model: without it, the translation performance degrades on average by 2 BLEU score points on EN→DE (see Appendix C). Finally, we qualitatively show how translation improves along the sampling steps by elimination of incoherent/repeated tokens in Table 5 in Appendix E.

## 3.2 TEXT GENERATION

**Unconditional Text Generation (Qualitative).** We train our method on a large high-quality publicly available Colossal Clean Common Crawl (C4) dataset (Raffel et al., 2019) to demonstrate samples. We tokenize the data using SentencePiece (Kudo & Richardson, 2018) [5] with vocabulary size 32K and train on randomly cropped sequences of length 32. The network is the same as the Transformer decoder part for MT in Section 3.1 but is much larger — 335M parameters: 24 layers,

---

[3]As shown in Table 7 in Appendix G, allowing larger compute reduces the gap even further, to $\triangle = 0.73$.

[4]For distillation, we follow the protocol used by Saharia et al. (2020). Scores in Table 1 were obtained with distillation from Transformer–Big to allow direct comparisons with CMLM (Ghazvininejad et al., 2019) and SMART Ghazvininejad et al. (2020); for Transformer–Base–distilled scores see Table 6 in Appendix F.

[5]https://github.com/google/sentencepiece

| Index | Type | Text |
|---|---|---|
| 1 | Prompt | Seeing * * * * * * for any visitor to a national park. While it is an exciting moment, a bear can * * * * * * * |
| | Completion | Seeing a big bear is a must for any visitor to a national park. While it is an exciting moment, a bear can be intimidating as it will come |
| 2 | Prompt | Seeing * * * * * * for any visitor to a national park. While it is an exciting moment, a moose can * * * * * * |
| | Completion | Seeing moose is a unique experience for any visitor to a national park. While it is an exciting moment, a moose can navigate its way through a forest |

Table 3: Inpainting from our model trained on C4 (cherry-picked).

1024 embedding size, 4096 hidden size, 16 attention heads. Same as in the MT setup, we remove the causal mask since our model is non-autoregressive. We train up to 400K steps with Adam optimizer with batch size 4096 and use cosine annealing for scheduling the learning rate with minimum $10^{-5}$, maximum $2 * 10^{-3}$ and linear warm-up of 10K steps from starting value of $10^{-7}$.

The trained model is then used for unconditional sampling. Starting with 32 random tokens, we iteratively perform up to 1K steps from our model, stopping earlier if samples do not change anymore. To make convergence faster, we use temperature 0.8 and update only 30% of all tokens at each step, chosen randomly — as explained in Section 2.2. We show 10 non-cherry-picked samples from our model in Table 8 of Appendix H. Of those samples, all but one resemble reasonable internet texts.

**Unconditional Text Generation (Quantitative).** While C4 is one of the largest high-quality datasets publicly available, it does not have an established benchmark among non-AR methods. To quantitatively evaluate our algorithm, we train it on the EMNLP2017 News[6] dataset, which has numerous text GAN baselines, including a strong ScratchGAN method (d'Autume et al., 2019). We also compare to several other baselines reported in (d'Autume et al., 2019; Caccia et al., 2020): SeqGAN (Yu et al., 2017), LeakGAN (Guo et al., 2017), RankGAN (Lin et al., 2017), MaliGAN (Che et al., 2017) and autoregressive (AR) language models.

We use the same architecture and optimizer as in qualitative experiments and train with batch size 1024 for 8K steps (chosen to achieve the lowest validation loss). We follow the same tokenization strategy as d'Autume et al. (2019), with vocabulary size of 5.7K and maximum length 52 tokens, padding shorter sequences to this maximum length. During sampling, we perform 1K steps and update all the tokens at each step (which we found beneficial for the chosen metrics). The quality/-variation trade-off is controlled by changing the sampling temperature in the range $[0.1, 2.0]$, using 60 values overall. For every temperature, we sample 10K texts for evaluating quality and another 10K texts for evaluating diversity, as done by d'Autume et al. (2019).

The results are shown in Figure 2. SUNDAE demonstrates good quality/variation trade-off, as measured by BLEU/self-BLEU metrics, outperforming 4 out of 5 GAN baselines, and competing with ScratchGAN and AR model. In the higher quality region, SUNDAE outperforms ScratchGAN, while in higher diversity region it slightly underperforms. The advantage of ScratchGAN in higher diversity region could be explained by its autoregressive decoding (it is a hybrid between AR and GAN). Interestingly, the curve for SUNDAE has a more complicated shape than the baselines — possibly because of its Markov Chain sampling, which can empirically exhibit something akin to "phase transitions". We show samples from SUNDAE in Table 9 of Appendix I.

**Text In-painting.** The non-autoregressive nature of our model opens up possibilities beyond left-to-right prompted generation, such as filling in arbitrary blank patterns in a template. In this section, we qualitatively investigate this new capability, using the C4-trained model described in previous sections, and another one, trained on a dataset of Python code from GitHub. All the architecture/training settings remain the same, except we train twice longer: up to 800K steps. Sampling settings are also the same, except we update all the tokens at each step as diversity is less of an issue with stronger conditioning. We construct the code dataset by extracting files ending in `.py` from open source GitHub repositories with licenses that are one of `apache-2.0, mit, bsd-3-clause, bsd-2-clause, unlicense, cc0-1.0, isc, artistic-2.0`. We perform document level de-duplication checking for exact matches.

For the C4 dataset, we present a model with a particular type of prompt `[CTX1][MSK1][CTX2][MSK2]`, where `[CTX]` stands for "context" and `[MSK]` for "masked", in Table 3, which shows how information can be "teleported" from arbitrary location to another arbitrary location, while taking into account bidirectional context. We change a species of the animal in `[CTX2]` and observe that the model actually uses the right word while in-painting

---

[6]http://www.statmt.org/wmt17/

| Index | Type | Text |
|---|---|---|
| 1 | Prompt | ```def trunc_len(x, t):  * * * * * * * * * * * * * * return min(n, t)``` |
| | Completion | ```def trunc_len(x, t):```
```    n = len(x)```
```    return min(n, t)``` |
| 2 | Prompt | ```is_even = lambda x:  * * * * * *```
```is_odd = lambda x:  not * * * * * * # reuse``` |
| | Completion | ```is_even = lambda x:  x % 2 == 0```
```is_odd = lambda x:  not is_even(x) # reuse``` |

Table 4: Inpainting from our model trained on GitHub (cherry-picked).

[MSK1] and plausibly continues [MSK2] according to the species. This kind of task would be impossible to perform for an AR model: with left-to-right causality it would be unable to correctly guess the name of the animal, while with a right-to-left one it would have to start filling [MSK2] tokens first, so it would just ignore the conditioning. By contrast, our model succeeds in this task.

For GitHub Python data, we use two different kinds of prompts for our model in Table 4: [CTX1][MSK1][CTX2] and [CTX1][MSK1][CTX2][MSK2][CTX3]. In the first example, our model guesses that n should be the length of the input sequence and correctly in-paints [MSK1] region. This would again be impossible to do for AR models. Looking from left-to-right, it is impossible to guess that n will be the name of the length variable and t is meant to be a threshold. Looking from right-to-left, it is unclear what variables n and t represent and what is the function being computed. Only a bidirectional model can perform well in this task. In the second example, our model correctly follows the suggestion to reuse the function which it also has to in-paint.

## 4 RELATED WORK

In this section we provide an overview of generative models, focusing on applications in text domain.

### 4.1 AUTOREGRESSIVE MODELS

Early attempts to perform autoregressive modelling of text using neural networks began with Bengio et al. (2003); later Sutskever et al. (2011) extended this approach with RNNs. Since then, improvements in neural network architecture, such as the Transformer (Vaswani et al., 2017), and scaling up as with GPT-3 (Brown et al., 2020) dramatically improved the performance of AR models. While the architectures evolved, the loss function remained the same with minor modifications.

Insertion (Stern et al., 2019) and Levenshtein (Gu et al., 2019) Transformers are deviations from the standard paradigm of training and could potentially overcome some of AR problems. However, in the parallel decoding regime, they have to rely on AR distillation for obtaining competitive results.

### 4.2 NON-AUTOREGRESSIVE MODELS IN GENERAL

Non-autoregressive models evolved in parallel with autoregressive ones but have so far struggled to address the same spectrum of tasks with the same quality as AR models.

**Diffusion** for generative models was originally introduced by Sohl-Dickstein et al. (2015). Recently Ho et al. (2020) proposed a different parametrization: instead of modeling one-step transitions, one could instead model $p(x_0|x_t)$ and later convert those to $p(x_{t-1}|x_t)$ by probabilistic reasoning. This was recently up-scaled for image modeling by Nichol & Dhariwal (2021) and extended to continuous time by Song et al. (2020). In terms of applications to text generation, two recent works addressed this: Hoogeboom et al. (2021) and Austin et al. (2021). Likelihoods obtained in those works are promising, but still behind AR and lacking sample quality as well.

Some diffusion works like Mittal et al. (2021) also approached modeling discrete sequences in another way: first encode sequences with a VAE into a continuous space and then apply continuous diffusion approaches commonly used for image generation.

**Variational Autoencoders (VAEs)** were proposed by Rezende et al. (2014) and Kingma & Welling (2013) as a likelihood-based extension of autoencoders. Application of such methods to text has been problematic empirically: as it was first noted in Bowman et al. (2015), good results come from using a strong AR decoder instead of a simple factorised one, but the latents are of-

ten ignored — which was named "posterior collapse problem". This problem was later tackled by Delta-VAEs (Razavi et al., 2019), with limited success in the text domain (Bosc & Vincent, 2020).

**Normalising Flows** (Dinh et al., 2014; Rezende & Mohamed, 2015; Dinh et al., 2016) are another prominent family of generative models, originally proposed for real-valued data and recently revisited for text generation by Hoogeboom et al. (2021). While conceptually interesting, the text samples of such methods lack in fidelity.

**Generative Adversarial Networks (GANs)** were originally introduced by Goodfellow et al. (2014) and remain one of the dominant methods for image generation. They were later adapted to text generation in Yu et al. (2017); Guo et al. (2017); Lin et al. (2017); Che et al. (2017). However, text generation with such methods is still a challenge, with a more recent ScratchGAN (d'Autume et al., 2019) showing relatively low-quality samples. At least part of the problem with applying GANs to text generation comes from non-differentiability of discrete text samples, which requires usage of zeroth order optimization methods.

**Energy-Based models** have a long history dating back to Hopfield (1982); Hinton (2002); LeCun et al. (2006); Ranzato et al. (2007). Recently, Deng et al. (2020) shows promising results for text.

### 4.3 NON-AUTOREGRESSIVE MODELS FOR MACHINE TRANSLATION

There have been many attempts to apply non-autoregressive methods to machine translation. Latent transformer (Kaiser et al., 2018) generated latent variables autoregressively and then decoded feed-forwardly. NAT (Gu et al., 2017) was the first to characterize a problem with non-autoregressive generation — "multi-modality". FlowSeq (Ma et al., 2019) applied Normalising Flows to machine translation. LVM-DAEs (Lee et al., 2018) are perhaps most related to our method, but these models do not have *unrolled denoising* and their decoding method is deterministic. The leading methods in this area are CMLM (Ghazvininejad et al., 2019) and DisCo (Kasai et al., 2020). While conceptually similar to LVM-DAEs (Lee et al., 2018), these works have introduced significant improvements to the training and decoding procedure, and, as a result, shown strong competition to AR methods. Both of them also do not have *unrolled denoising* and sampling like we do, but they are still powerful baselines. Later, there were attempts to combine CMLM with local AR (Kong et al., 2020). Perhaps the most related to our work is the SMART (Ghazvininejad et al., 2020) follow-up of the CMLM. However, this method is not probabilistic and it does not show unconditional generation capabilities. Finally, a recent line of work called Imputer (Chan et al., 2020; Saharia et al., 2020) achieves strong results in machine translation by aligning target to source via dynamic programming.

### 4.4 DENOISING OBJECTIVE IN GENERAL

With the advent of BERT (Devlin et al., 2018), the denoising objective became very popular for text representation learning. The original idea was later simplified in RoBERTa (Liu et al., 2019) by removing the unnecessary next-sentence-prediction task and only retaining the denoising task (the cloze task). This is similar to our work, however, we do not apply masking to corrupt the input tokens, and instead only switch them to random ones. More importantly, RoBERTa does not use *unrolled denoising*. Another work, Electra (Clark et al., 2020a), deals with the same kind of corruption that we use, but instead of predicting the original token before the corruption, Electra predicts whether it was corrupted or not. Electric (Clark et al., 2020b) does not perform masking, but instead uses noise-contrastive loss to learn a representation of text. BART (Lewis et al., 2019) uses both BERT-style denoising and autoregressive modeling for learning representations. All of these methods were originally motivated by improving representation learning of text. There were a few attempts to heuristically decode models like BERT and turn them into text generative models such as in Wang & Cho (2019), however, samples from these models lack coherence.

## 5 CONCLUSION

In this paper we have proposed a novel non-autoregressive method that operates within the framework of denoising autoencoders and, crucially, unrolls the denoising process during training. *Unrolled denoising* allows us to achieve state-of-the art results in WMT'14 English-to-German translation task amongst non-AR methods (without distillation from large AR models) and good qualitative results in unconditional text modeling on C4 dataset. We have also qualitatively demonstrated new inpainting capabilities on C4 and GitHub Python data, which opens up new avenues for creative text editing where a human could more naturally collaborate with a machine on writing and even programming.

## AUTHOR CONTRIBUTIONS

Nikolay Savinov came up with the idea of unrolled denoising, wrote the first prototype of unconditional generation, contributed to the machine translation codebase and co-led the project. Junyoung Chung wrote most of the machine translation codebase, came up with the length prediction conditioning, suggested argmax-unrolled decoding and co-led the project. Mikołaj Bińkowski came up with theoretical insights about our model, wrote most of the evaluation codebase, implemented model improvements and co-led the project. All three first authors contributed equally to writing the paper. Erich Elsen gave high-level scientific guidance, suggested text in-painting experiments and made substantial edits to the paper draft. Aäron van den Oord originally suggested to look into denoising generative models for text, gave high-level scientific guidance, proposed machine translation experiments and made substantial edits to the paper draft.

## ACKNOWLEDGMENTS

We would like to thank Jean-Baptiste Alayrac and Miruna Pîslar for valuable discussions on the machine translation experiments, William Chan and Chitwan Saharia for sharing distilled translation datasets, Mihaela Rosca and Cyprien de Masson d'Autume for sharing text GAN evaluation code, Yujia Li and Jack Rae for sharing a dataset of Python code from GitHub, Sander Dieleman and Oriol Vinyals for in-depth feedback about our work, Jungo Kasai for responses to our questions about DisCo, Dani Yogatama for sharing knowledge on machine translation evaluation, and Jeff Donahue for feedback and discussions.

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

## A   METHOD DETAILS

Using Markov property and the form of $p_t = f_\theta(\cdot|\mathbf{x}_{t-1})$, for any $t > 0$ we obtain

$$p_t(\mathbf{x}_t|\mathbf{x}_0) = \sum_{\boldsymbol{x}_{t-1} \in X} p_{t-1}(\boldsymbol{x}_{t-1}|\mathbf{x}_0) f_\theta(\mathbf{x}_t|\boldsymbol{x}_{t-1}) \tag{5}$$

$$= \mathbb{E}_{\mathbf{x}_{t-1} \sim p_{t-1}(\cdot|\mathbf{x}_0)} f_\theta(\mathbf{x}_t|\mathbf{x}_{t-1}). \tag{6}$$

By induction, we can keep unrolling the chain for all steps $s = t - 1, t - 2, \ldots, 1$, eventually obtaining

$$p_t(\boldsymbol{x}_t|\boldsymbol{x}_0) = \sum_{\substack{\boldsymbol{x}_1,\ldots,\boldsymbol{x}_{t-1} \\ \in X}} \prod_{s=1}^{t} f_\theta(\boldsymbol{x}_s|\boldsymbol{x}_{s-1})$$

$$= \mathbb{E}_{\substack{\mathbf{x}_1 \sim p_1(\cdot|\boldsymbol{x}_0) \\ \cdots \\ \mathbf{x}_{t-1} \sim p_{t-1}(\cdot|\mathbf{x}_{t-2})}} [f_\theta(\boldsymbol{x}_t|\mathbf{x}_{t-1})], \tag{7}$$

which proves the from of the model distribution (Eq. 1).

### A.1   UPPER BOUND ON THE LOSS.

Using Jensen's inequality we can swap $\log$ and expectation in Eq. 7 to obtain

$$-\log p_t(\mathbf{x}|\mathbf{x}_0) \leq -\mathbb{E}_{\substack{\mathbf{x}_1 \sim p_1(\cdot|\boldsymbol{x}_0) \\ \cdots \\ \mathbf{x}_{t-1} \sim p(\cdot|\mathbf{x}_{t-2})}} [\log f_\theta(\mathbf{x}|\mathbf{x}_{t-1})], \tag{8}$$

hence

$$\widetilde{L^{(t)}}(\theta) = -\mathbb{E}_{\substack{\mathbf{x} \sim p_{\text{data}} \\ \mathbf{x}_0 \sim q(\cdot|\mathbf{x})}} [\log p_t(\mathbf{x}|\mathbf{x}_0)] \leq -\mathbb{E}_{\substack{\mathbf{x} \sim p_{\text{data}} \\ \mathbf{x}_0 \sim q(\cdot|\mathbf{x}) \\ \mathbf{x}_1 \sim f_\theta(\cdot|\mathbf{x}_0) \\ \cdots \\ \mathbf{x}_{t-1} \sim f_\theta(\cdot|\mathbf{x}_{t-2})}} [\log f_\theta(\mathbf{x}|\mathbf{x}_{t-1})] = L^{(t)}(\theta). \tag{9}$$

### A.2   RELATION TO VAE

Variational Autoencoders (Kingma & Welling, 2013; Rezende et al., 2014) optimise the expected evidence lower bound (ELBO) on the marginal likelihood

$$\log p(\mathbf{x}) \geq -D_{\text{KL}}(q(\mathbf{z}|\mathbf{x})\|p_0(\mathbf{z})) + \mathbb{E}_{\mathbf{z} \sim q(\cdot|\mathbf{x})} [\log p(\mathbf{x}|\mathbf{z})], \tag{10}$$

with respect to parameters of both the variational distribution $q$ and generative distribution $p$, where $p_0$ remains the prior for for the latent variables. Denosing autoencoders limit the optimisation to the second term, assuming a fixed encoder $q$, which is also the case with SUNDAE: identifying encoder with our corruption distribution and generative distribution $p$ with the distribution of our chain at the last step $p_T$, expected ELBO becomes

$$\mathbb{E}_{\mathbf{x} \sim p_{\text{data}}}[\log p_T(\mathbf{x})] \geq -\mathbb{E}_{\mathbf{x} \sim p_{\text{data}}} [D_{\text{KL}}(q(\mathbf{z}|\mathbf{x})\|p_0(\mathbf{z}))] + \mathbb{E}_{\substack{\mathbf{x} \sim p_{\text{data}} \\ \mathbf{x}^c \sim q(\cdot|\mathbf{x})}} [\log p_t(\mathbf{x}|\mathbf{x}^c)]$$

$$\geq -\mathbb{E}_{\mathbf{x} \sim p_{\text{data}}} [D_{\text{KL}}(q(\mathbf{z}|\mathbf{x})\|p_0(\mathbf{z}))] - L^{(T)}(\theta). \tag{11}$$

where the second inequality is a consequence of the upper bound 3. Hence, minimizing $L^{(T)}$ would lead to maximisation of the lower bound on the model likelihood.

### A.3   CORRUPTION FUNCTION

Our corruption function is obtained in stages:

- sample uniform expected corruption proportion $\alpha \sim \mathcal{U}([0, 1])$,
- sample Bernoulli mask, independently for each token $\boldsymbol{m} \sim (\text{Bernoulli}(\alpha))^N$,
- sample noise $\eta \sim (\mathcal{U}(\{1, 2, \ldots, v\}))^N$,

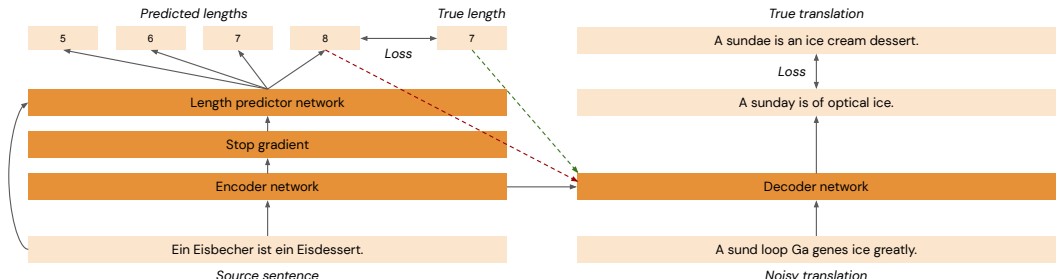

Figure 3: Overview of target length prediction. During SUNDAE training, we simultaneously train the length predictor with cross-entropy loss and give ground-truth target length as input to the decoder (green dashed arrow), thus teacher-forcing it. During sampling, we give the most likely target length prediction from the network to the decoder (red dashed arrow).

- compute corruption $\boldsymbol{x}^c = c_{\alpha,\boldsymbol{m},\eta}(\boldsymbol{x}) = (1 - \boldsymbol{m}) \cdot \boldsymbol{x} + \boldsymbol{m} \cdot \eta$ (with multiplications taken element-wise).

The distribution $p^c(\cdot|\boldsymbol{x})p_{\text{data}}(\boldsymbol{x})$ obtained in this process covers the entire input space $X$, with probability skewed towards samples whose parts come from $p_{\text{data}}$. Note that except for corruption proportion, all steps are independent for all individual tokens. Given $\alpha$, $p^{c_\alpha}(\boldsymbol{y}|\boldsymbol{x})$ can be factorised as

$$\prod_n p^{c_\alpha}(\boldsymbol{y}^{(n)}|\boldsymbol{x}^{(n)}) = \prod_n h(\boldsymbol{y}^{(n)})^T \boldsymbol{Q}_\alpha h(\boldsymbol{x}^{(n)}),$$

where $\boldsymbol{Q}_\alpha = \alpha\boldsymbol{I} + (1 - \alpha)\boldsymbol{V}$ is a transition matrix, $\boldsymbol{V}$ is a $v \times v$ matrix with all entries equal to $\frac{1}{v}$, and $h(a)$ denotes one-hot vector representation of $a \in \{1, 2, \ldots, v\}$. In this context, corruption with rate $\alpha$ corresponds to a *multinomial-diffusion forward process* with probability $\alpha$ of remaining in the same state, as proposed by (Hoogeboom et al., 2021).

## B    DETAILS OF TARGET LENGTH PREDICTION

We provide a simplified high-level overview of the target length predictor in Figure 3. In the rest of this section we focus on implementation details.

The target length prediction module consists of 6 residual blocks. The length prediction module takes the source encodings $\boldsymbol{h_x} = encoder(\boldsymbol{x})$ as the input, where $\boldsymbol{x}$ is a sequence of source tokens. We first project $\boldsymbol{h_x}$ into a vector $\boldsymbol{v_x} \in \mathbb{R}^{d_{LP}}$, where $d_{LP} = 128$ is the hidden size of the target length prediction module. We add a source length embedding vector $\boldsymbol{v}_{len} = S_{(N_{source} \times d_{LP})}(l_{\boldsymbol{x}})$ to $\boldsymbol{v_x}$ and obtain the final encoder state representation $\boldsymbol{v}'_{\boldsymbol{x}} = \boldsymbol{v_x} + \boldsymbol{v}_{len}$, where $l_{\boldsymbol{x}}$ is the number of the source tokens, and $N_{source} = 128$ is the maximum number of the source tokens. The target length prediction module then takes $\boldsymbol{v_x}$ as the input and predicts the (downsampled) target length $\tilde{l}_d$. We found beneficial downsampling the target lengths for the length prediction by a factor of 2, i.e. $l_d = \lceil l/2 \rceil$ as compared to exact prediction of $l$; since the maximum number of target tokens is $N = 128$, the maximum size of the outcome of length prediction is $N_d = 64$.

We prepend an embedding $\boldsymbol{h}_{len} \in \mathbb{R}^{d_{enc}}$ of target length to the source encodings $\boldsymbol{h_x} \in \mathbb{R}^{N_{source} \times d_{enc}}$ for each source example $\boldsymbol{x}$ and allow the decoder to attend to it ($d_{enc}$ here denotes the dimensionality of the encoder). $\boldsymbol{h}_{len}$ is obtained by applying an embedding matrix $V_{(N_d \times d_{enc})}$ to ground truth (downsampled) length $l_d$ at training time, or to predicted one $\tilde{l}_d$ during sampling. Note that optimisation of the length classification loss does not affect the encoder parameters.

## C    EFFECTS OF TARGET LENGTH PREDICTION

It is shown empirically that the length of the translated results can affect BLEU score. Approximate search methods such as beam search usually incorporate a length normalization term in order to prevent the search algorithm to prefer shorter sentences. This is due to the fact that the scoring function

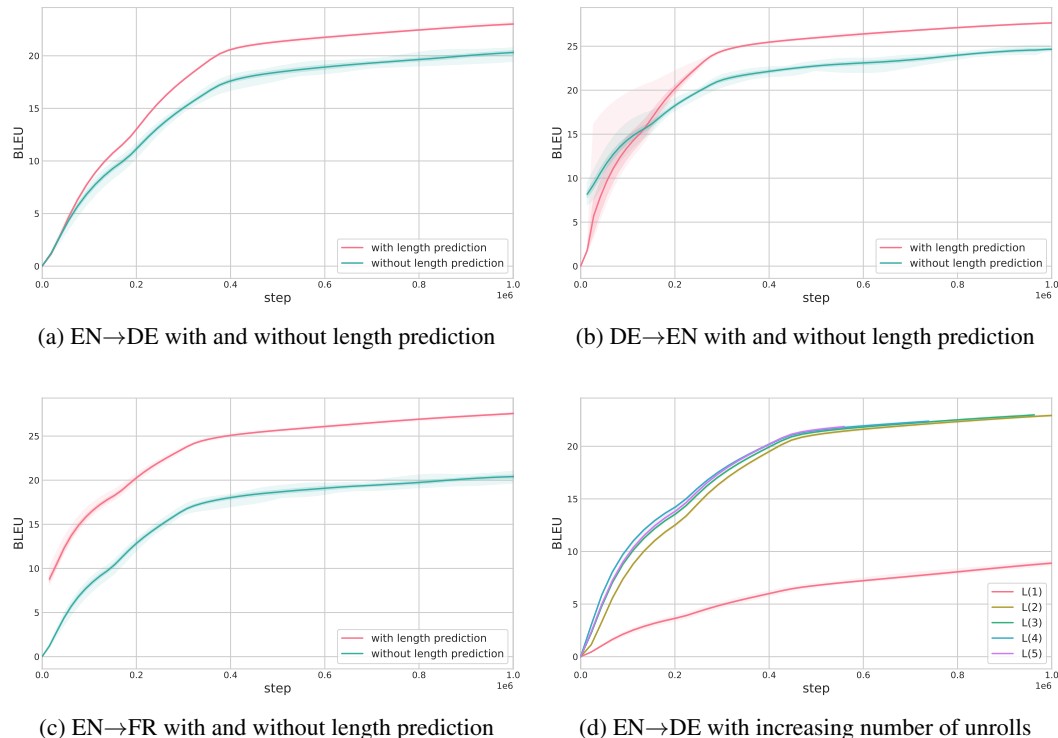

(a) EN→DE with and without length prediction

(b) DE→EN with and without length prediction

(c) EN→FR with and without length prediction

(d) EN→DE with increasing number of unrolls

Figure 4: Validation BLEU curves during training with and without target length prediction on WMT'14 EN→DE, DE→EN and EN→FR tasks shown in Figure 4a, 4b and 4c, respectively. Figure 4d shows the effect of *unrolled denoising* terms on translation quality, here $L(k) = L^{(1:k)}$. All models are trained with a batch size of 256 using 20 different random seeds to display uncertainty.

| Source | Ich habe dort Dinge gesehen, aus denen ich gestärkt hervorgegangen bin. |
|---|---|
| Initialization | ThirVerschlMarta moderatopposed frameworks solidierweitert [...] |
| Step 1 | I saw things there that which I I stronger.. |
| Step 2 | I saw things there from which I me stronger.. |
| Step 3 | I saw things there from which I emerged stronger. |
| Step 4 | I saw things there from which I emerged stronger. |
| Reference | I saw some things there and came out stronger. |

Table 5: German-to-English translation process. Since initialization is quite long, we substitute the trailing tokens with [...]. Tokens changed from previous step are highlighted in gray. The process converges after 3 steps, while AR would take 10 steps (one for each token in the translation).

is usually chosen as the likelihood modelled by the translation system, and beams containing shorter sequences tend to score higher likelihoods. In non-AR models, the end of sequence is not explicitly learned by the models, thus they can benefit by knowing the target sequence length in advance to generating the translation. However, the length of the target sequence cannot be known at inference time, therefore, a model should predict it instead. We show how BLEU scores can differ in WMT'14 translation tasks with and without the target length prediction in Figure 4a-4c.

## D    EFFECTS OF UNROLLED LOGITS LOSSES

We show how varying the number of *unrolled logits* can affect the MT performance in Figure 4d.

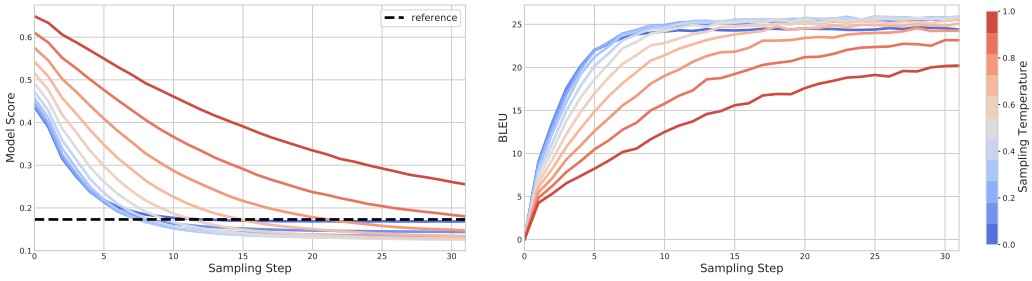

(a) Model score evolution throughout sampling.          (b) BLEU evolution throughout sampling.

Figure 5: Model score (left) and BLEU (right) evolution throughout sampling for various temperatures. The reference score on the left hand side denotes the model score obtained for ground truth inputs. Overall, medium-low temperatures give the best results.

## E   TRANSLATION ANALYSIS

We show how our algorithm works step-by-step for translation in Table 5. It is interesting to observe how the multimodality problem (Gu et al., 2017) gets resolved with more steps. Multimodality usually manifests in repeated tokens in translation, which intrinsically comes from inability to coordinate decisions when independently sampling multiple tokens at once. However, after a few steps all repetitions are corrected, because the model conditions on the previous step and hence can better coordinate further changes.

Figure 5 shows how the model and the BLEU scores evolve during sampling for various temperatures. The model score here refers to the cross-entropy between translation and model logits produced by giving this translation as the input. While for very low temperatures the scores initially improve faster, they are eventually outperformed by scores for higher temperatures, possibly due to low diversity induced by near-deterministic sampling. For very high temperatures, on the other hand, the scores improve slowly, and hence using such temperatures is rather impractical.

## F   DISTILLATION SCORES OBTAINED WITH TRANSFORMER–BASE

In Table 6 we also present results for SUNDAE trained on data distilled from autoregressive Transformer-Base model, which allows direct comparison with similarly obtained Imputer scores (Saharia et al., 2020). The results show that SUNDAE noticeably outperforms Imputer at 8 steps for both EN→DE and DE→EN language pairs.

| | | AR-distilled BLEU | |
| Model | Steps ($T$) | EN→DE | DE→EN |
| --- | --- | --- | --- |
| Imputer (Saharia et al., 2020) ($n{=}1$) | 4 | 27.9 | 30.9 |
| Imputer (Saharia et al., 2020) ($n{=}1$) | 8 | 27.9 | 31.1 |
| **SUNDAE (63M)** | | | |
| Deterministic ($n{=}16$) | 4 | 28.00 | 31.82 |
| Deterministic ($n{=}16$) | 8 | 28.01 | 31.84 |
| Deterministic ($n{=}16$) | 10 | 28.01 | 31.84 |
| Stochastic ($n{=}16$) | 4 | 27.78 | 31.42 |
| Stochastic ($n{=}16$) | 8 | **28.14** | 31.78 |
| Stochastic ($n{=}16$) | 10 | 28.11 | 31.87 |
| Stochastic ($n{=}16$) | 16 | 28.11 | 31.87 |
| Stochastic ($n{=}16$) | 32 | 28.02 | **31.92** |

Table 6: Test BLEU scores of SUNDAE and Imputer (Saharia et al., 2020) distilled from AR Transformer–Base model on English-to-German (EN→DE) and German-to-English (DE→EN) translation tasks.

## G  SACREBLEU SCORES FOR WMT'14 EXPERIMENTS

SacreBLEU (Post, 2018) is a library for computing BLEU score[7] that does not require the user to manually tokenize the reference and candidate translations. We report BLEU scores measured with SacreBLEU library in order to provide a reference for future studies. We show all the BLEU scores in Table 7.

| Model | Steps ($T$) | EN→DE | | DE→EN | | EN→FR | |
|---|---|---|---|---|---|---|---|
| | | BLEU | BLEU$^\star$ | BLEU | BLEU$^\star$ | BLEU | BLEU$^\star$ |
| Deterministic ($n=16$) | 2 | 20.29 | 20.13 | 25.28 | 24.83 | 31.60 | 30.51 |
| Deterministic ($n=16$) | 3 | 24.07 | 23.84 | 28.64 | 28.12 | 35.88 | 34.68 |
| Deterministic ($n=16$) | 4 | 25.01 | 24.75 | 29.53 | 28.99 | 36.85 | 35.61 |
| Deterministic ($n=16$) | 10 | 25.54 | 25.25 | 30.11 | 29.54 | 37.15 | 35.92 |
| Stochastic ($n=16$) | 4 | 23.05 | 22.75 | 28.13 | 27.62 | 35.27 | 34.03 |
| Stochastic ($n=16$) | 8 | 26.22 | 26.08 | 30.48 | 30.00 | 37.45 | 36.16 |
| Stochastic ($n=16$) | 10 | 26.25 | 25.99 | **30.80** | **30.24** | 27.53 | **36.23** |
| Stochastic ($n=16$) | 32 | **26.57** | **26.31** | 30.74 | 30.11 | **37.60** | 36.20 |

Table 7: Test BLEU without AR distillation (raw). Scores computed with SacreBLEU library are denoted as BLEU$^\star$.

## H  UNCONDITIONAL SAMPLES FOR C4

We provide samples for our model trained on C4 dataset as described in the main text (Section 3.2). All of them resemble reasonable-quality internet texts, except perhaps sample #4.

| Index | Sample from our model |
|---|---|
| 1 | I had to go back and forth and start over to see what had happened, what was working, and all the pieces were in the right places. |
| 2 | tossed in a delicious sauce. Beef brisket is a hearty cut of meat – thick and tender. When you make a lot of meat |
| 3 | can expect to see more of the same in 'Wild Herring', although not very often enough to achieve the same effect. In 'Herring', |
| 4 | , she was considered to be a fake. Dr. Deborah Chienna has 93 in the last 365 days., headaches, stomach pain, sore as an artist. |
| 5 | printing photo cards, artwork, and posters. email me for some ideas. "Rebecca's studio |
| 6 | Jersey, Philadelphia and Washington DC. This is a conference attended by residents, physicians, graduate medical students, and others in the field of public health. |
| 7 | 's location. This research was supported by a grant from the National Association for the Advancement of Science through the U.S. Department of Energy |
| 8 | different artists, we would love to show you the piece we have in our shop! Keep an eye on our Gallery – all you have to do is |
| 9 | 6.3 million from $88.8 million reported by Zacks.com back in February. Brad Brickley, general manager of Irvine, Calif.-based |
| 10 | to writing with the Autumns. Together, some of the class will spend time in the classroom to read a poem from the original book. From there we |

Table 8: Unconditional samples from our model trained on C4 (without cherry-picking). Since C4 was crawled from the web, newline symbols are abundant both in the training data and the samples.

---

[7]https://github.com/mjpost/sacrebleu

# I  UNCONDITIONAL SAMPLES FOR EMNLP2017 NEWS

We provide samples from our model trained on EMNLP2017 News dataset in Table 9, accompanying the quantitative results in the main text (Section 3.2). These samples are obtained at temperature 0.8. None of them appear in the training set, so the model does not merely memorize the data it is given. To provide a point of reference, we also include samples from ScratchGAN (d'Autume et al., 2019).

| Index | Sample |
|---|---|
| **ScratchGAN** | |
| 1 | We are pleased for the trust and it was incredible , our job quickly learn the shape and get on that way . |
| 2 | But I obviously have him with the guys , maybe in Melbourne , the players that weren ' t quite clear there . |
| 3 | There is task now that the UK will make for the society to seek secure enough government budget fund reduce the economy . |
| 4 | Keith is also held in 2005 and Ted ' s a successful campaign spokeswoman for students and a young brothers has took an advantage of operator . |
| 5 | Police said how a Democratic police officer , would choose the honor of alcohol and reduce his defense and foundation . |
| 6 | We do not go the Blues because that I spent in ten months and so I didn ' t have a great job in a big revolution . |
| 7 | The 28 - year - old - son Dr Price said she would have been invited to Britain for her " friend " in a lovely family . |
| 8 | And as long as it is lower about , our families are coming from a friend of a family . |
| **SUNDAE** | |
| 1 | Ms Sturgeon was quoted as saying that she is prepared to quit the EU by the end of March next year . |
| 2 | They ' ve been around a long time , but it ' s too natural to ignore what has happened . |
| 3 | " It ' s a busy road , and we ' ve got to get through it , " he said . |
| 4 | That means some voters will decide if they ' ll change their minds before the first day of the debate . |
| 5 | " We don ' t learn very much from my point of view because we live here , " he said . |
| 6 | " I spent my whole life at the stage and now I ' m a normal father , " he continued . |
| 7 | Whether your phone is near or online , you can check if your phone is tied to your Instagram account . |
| 8 | " It is only too early to know why the incident occurred shortly after it happened , " he told the Associated Press . |
| 9 | The website will be updated on a day - to - day basis , as well as other special services . |
| 10 | So it ' s too early to know exactly what each candidate has to say in terms of the American election . |

Table 9: Unconditional samples from our model SUNDAE trained on EMNLP2017 News (without cherry-picking). We also provide samples from ScratchGAN (d'Autume et al., 2019) for comparison.

# J  PSEUDOCODE OF SUNDAE

We provide Python-like pseudocode for our method in Listing 1. The main functions are `build_loss_fn`, which should be used for training, and `sampling_fn` which should be used for sampling. For simplicity, we consider the decoder-only setup as in unconditional generation experiments (Section 3.2).

```
1  def get_random_text(shape, vocab_size):
2    random_text = rand_int(shape, minval=0, maxval=vocab_size)
3    return random_text
4
5
6  def corrupt_text(batched_text, vocab_size):
7    corruption_prob_per_sequence = rand_uniform([batched_text.shape[0], 1])
8    rand = rand_uniform(batched_text.shape)
9    mask = rand < corruption_prob_per_sequence
10   random_text = get_random_text(batched_text.shape, vocab_size)
11   corrupted = mask * random_text + (1 - mask) * batched_text
12   return corrupted
13
14
15 def build_logits_fn(vocab_size, n_unrolled_steps, enable_sampling):
16
17   def logits_fn(batched_text):
18     model = Transformer(vocab_size, use_causal_mask=False)
19
20     def fn(input_batched_text):
21       logits = model(input_batched_text)
22       return logits
23
24     def unrolled_fn(input_batched_text):
25       samples = corrupt_text(input_batched_text, vocab_size)
26       all_logits = []
27       for _ in range(n_unrolled_steps):
```

```python
28          logits = fn(samples)
29          samples = stop_grad(rand_categorical(logits))
30          all_logits += [logits]
31        final_logits = concatenate(all_logits, axis=0)
32        return final_logits
33
34      if enable_sampling:
35        return fn(batched_text)
36      else:
37        return unrolled_fn(batched_text)
38
39    return logits_fn
40
41
42  def build_loss_fn(vocab_size, n_unrolled_steps=2):
43    logits_fn = build_logits_fn(
44        vocab_size, n_unrolled_steps, enable_sampling=False)
45
46    def loss_fn(batched_text):
47      logits = logits_fn(batched_text)
48      # repeat batched_text to fit unrolled logits
49      targets = concatenate([batched_text] * n_unrolled_steps, axis=0)
50      one_hot_targets = one_hot(targets, vocab_size)
51      loss_per_token = -sum(
52          one_hot_targets * log_softmax(logits), axis=-1)
53      loss = mean(loss_per_token)
54      return loss
55
56    return loss_fn
57
58
59  def sampling_fn(logits_fn, steps, temperature, batch_size,
60                  sequence_length, vocab_size):
61    batched_text = get_random_text(
62        shape=[batch_size, sequence_length], vocab_size)
63    for _ in range(steps):
64      logits = logits_fn(batched_text)
65      samples = rand_categorical(logits / temperature)
66      batched_text = samples
67    return batched_text
```

Listing 1: Pseudocode of SUNDAE.

