# OpenReview forum: "Step-unrolled Denoising Autoencoders for Text Generation"
_ICLR.cc/2022/Conference — ICLR 2022 Poster_

### Official Review · Reviewer_D7fL · 2021-10-31

**Correctness:** 4
**Technical Novelty And Significance:** 4
**Empirical Novelty And Significance:** 3
**Recommendation:** 6
**Confidence:** 3

**Main Review:**

The proposed method is well-motivated and looks reasonable: it provides better noisy samples in addition to the random corrupted ones for training the denoising autoencoder. It has promising performance on the EN->DE translation compared to several baselines. The effectiveness is confirmed by ablation studies.

Besides, my concerns have two-fold.

One is regarding the evaluation in this paper.
(1) I found the table that reports the main results is remarkably sparse. Although it lists a great number of baselines in comparison, almost all these baselines were evaluated only on AR-distilled BLEU, whereas the proposed models were only evaluated with Raw BLEU. This makes the many scores in Table 2 do not comparable. For baselines that have Raw BLEU score, only one baseline is available for DE->EN translation;
(2) The results of the unconditional text generation experiment have never been quantitatively evaluated. This makes me feel that this experiment is only for illustrating "how can the proposed model apply to language modeling tasks" rather than "how well is the proposed model on language modeling tasks".
(3) In the experiment section, the authors only talk about what they have done and what they have seen from the table. More explanations, interpretations, and in-depth analyses are welcome.
(4) As the evaluation has only been done on only MT on a single language pair, it would be interesting to see how and how well the proposed model handles other language generation tasks. I personally view that MT is not a typical LG task as its outputs highly depend on its inputs. I wonder, for tasks that require planning and require the generated text to be diverse (in line with most text generation tasks), will such a denoising technique still work?

The other is that I found the present paper is a bit hard to follow. This is probably because, for example, variables are never defined before use, and, for example, many terms (e.g., step) refer to multiple concepts but sometimes can not be disambiguated given their contexts.

**Summary Of The Paper:**

The present paper proposes a Step-untolled denoising autoencoder (SUNDAE), a NAR text generation model which marries de-noising auto-encoder and Markov chain models. The major contribution of this paper is the unrolled denoising training scheme, which is a training step method. In the first step, the original text is randomly corrupted in the same way as what the denoising autoencoder did. Subsequently, in the second step, the noisy text from the first step is first denoised by sampling from the model, and the denoised text is then used as a sample for training the denoising autoencoder.

The proposed model has been tested on two tasks that involve text generation: machine translation as well as language modeling.

**Summary Of The Review:**

The proposed model is interesting, reasonable, and well-motivated. It has promising performance in ablation studies. Nevertheless, the evaluation part has clear flaws and the paper itself is sometimes hard to follow.

---

> ### Author Response · Authors · 2021-11-20
> **Authors' response**
>
> We thank the reviewer for their work on the review.
>
> __Add AR-distilled BLEU:__
> Addressed in the message ["Authors' response to all reviewers"](https://openreview.net/forum?id=T0GpzBQ1Fg6&noteId=WkxsAkcWC3V), please see above.
>
> __In the experiment section, the authors only talk about what they have done and what they have seen from the table. More explanations, interpretations, and in-depth analyses are welcome.__
> We have added additional in-depth analyses of our experimental results to the supplementary material (due to space limitations). In particular, we added step-by-step translation decoding visualization in Table 6 and analyzed it in Appendix E. We also plotted BLEU score evolution along the sampling steps in Figure 5 for multiple temperatures and also discussed it in Appendix E. Finally, we have ablations of our method in Figure 4, which we originally kept in supplementary for the space reasons. We are happy to add other kinds of analysis if some aspects of our work remain under-investigated.
>
> __The evaluation has only been done on MT on a single language pair:__
> We do have results on the En-Fr language pair in "Results and ablation studies", not only on En-De. SUNDAE achieved 37.77 BLEU on EN→FR, where AR reported 38.1 BLEU. Thus, SUNDAE is only 0.33 BLEU behind the standard AR baseline on EN→FR without distillation. We will make those results more prominent in the paper.
>
> __I wonder, for tasks that require planning and require the generated text to be diverse (in line with most text generation tasks), will such a denoising technique still work?__
> Addressed in the message ["Authors' response to all reviewers"](https://openreview.net/forum?id=T0GpzBQ1Fg6&noteId=WkxsAkcWC3V), please see above.
>
> __Paper is a bit hard to follow: variables are never defined before use, and, for example, many terms (e.g., step) refer to multiple concepts but sometimes can not be disambiguated given their contexts:__
> To improve clarity, we updated the method section: shortened it and moved less important parts to the supplementary. Please let us know if there is more we can do to make the paper easier to read.
>
> We have added extra experimental results that were mentioned by the reviewer to the manuscript. If your concerns are alleviated by these extra results, we respectfully ask the reviewer whether they would change their rating.

---

> > ### Comment · Reviewer_D7fL · 2021-11-24
> > **Response**
> >
> > > To summarize the results, we show significant improvement with respect to the baselines, in both EN -> DE and DE -> EN pairs, and set new SOTA among distillation-based non-AR results.
> >
> > Given the numbers in Table 1, I don't think there is a significant improvement in DE->EN and the proposed model set a new SOTA. The authors need to elaborate on why they said this.
> >
> > > That said, we believe non-distilled (raw) results originally presented in our paper are still valuable: distillation makes the training pipeline complicated and resource-consuming in practice because it requires sequentially decoding a pre-trained AR model on the entire training dataset (4.5M sentences).
> >
> > I agree, but, in Table 1, most baselines were not evaluated by raw BLEU.
> >
> > > I wonder, for tasks that require planning and require the generated text to be diverse (in line with most text generation tasks), will such a denoising technique still work? Addressed in the message "Authors' response to all reviewers", please see above.
> >
> > Thanks for adding more results on unconditional text generation, but I don't think this answer my question as unconditional text generation is not a ``real'' NLG task.
> >
> > Generally speaking, although the effectiveness of the model is still questioned, we can learn something interesting from the paper. I will not change my score.

---

> > > ### Author Response · Authors · 2021-11-24
> > > **Authors' response**
> > >
> > > Thanks for your quick response.
> > >
> > > __Given the numbers in Table 1, I don't think there is a significant improvement in DE->EN and the proposed model set a new SOTA. The authors need to elaborate on why they said this.__
> > > This is a communication error in our rebuttal, this should have been "We show significant improvement with respect to the baselines on EN -> DE", because the model doesn’t outperform Guo et al. on DE -> EN. We edited the wording in the message.
> > >
> > > __Thanks for adding more results on unconditional text generation, but I don't think this answer my question as unconditional text generation is not a ``real'' NLG task.__
> > > In this paper we focus on MT and unconditional generation as these are the extremes to show both that we can provide high-quality and diverse generations. Are there any specific tasks the reviewer wanted to see? Most of the other non-AR baselines that we compare to don't include other tasks.

---

### Official Review · Reviewer_7uVK · 2021-11-02

**Correctness:** 4
**Technical Novelty And Significance:** 2
**Empirical Novelty And Significance:** 2
**Recommendation:** 5
**Confidence:** 4

**Main Review:**

Non-AR generation is a hot topic recently, this paper proposes a new NAR method from the perspective of denoising autoencoder. The related works are well-organized and discussed thoroughly.

But the following flaws prevent me from giving a higher score.
-  The contribution seems incremental. The one-step denoising resembles the training of BERT and the related NAR method mask-predict (Ghazvininejad et al., 2019). And the two-step denoising, which is the main contribution of the paper, also resembles an improvement of mask-predict [1] which conducts one-step generation while training and takes the result as the model output. Can you compare it and claim the difference and the contribution of this paper?
- For NAR models, the decoding speed (or latency) is a crucial evaluation metric, which is missed in this paper. From Table 2, we can observe that the proposed method needs 32 decoding steps to obtain good results. In my experience, the average length of sentences in WMT14 En-De lies around 20-30, which means that the proposed method is slower than the AR counterpart in a majority of cases. Can you report the decoding speed in Table 2 to construct a complete comparison?
- Obtaining good results w/o kd is an advantage of the method. But I think it is worthy to utilize kd if it can provide better results. Does kd brings benefits to your method and why? KD alleviates the multi-modality problem in the dataset so that the NAR model can be more easily trained. Your method may also achieve a similar effect. Therefore I suggest authors add an investigation w.r.t the method's capacity of alleviating the multi-modality problem.
- As for the experiments of unconditional text generation, I suggest the authors provide quantitive evaluation scores to provide a clear comparison.


[1] Semi-Autoregressive Training Improves Mask-Predict Decoding, Marjan Ghazvininejad et.al., 2020

**Summary Of The Paper:**

The paper proposes an unrolled denoising method for non-autoregressive text generation. Specifically, the method first corrupts the original input and then re-generate uncertain tokens in it, and the result is taken as the output of the denoising autoencoder to compute loss with the target. Experiments are conducted on machine translation and unconditional text generation. The method achieves good results on MT without utilizing knowledge distillation.

**Summary Of The Review:**

The paper studies NAR generation, which is an interesting topic. The proposed method achieves good results, but the flaws on contribution/evaluation prevent me from giving a higher score.

---

> ### Author Response · Authors · 2021-11-20
> **Authors' response**
>
> We thank the reviewer for their work on the review.
>
> __Can you compare to SMART and claim the difference and the contribution of this paper?__
> Thank you for bringing SMART into our attention, it is indeed related but went under our radar previously. We have now added SMART to our main table and to the related work discussion. Comparison in distilled BLEU shows that SUNDAE outperforms SMART for the same number of decoding steps T=10: +0.46 BLEU for En->De and +0.6 BLEU for De->En. Raw BLEU is not reported in SMART. While SMART starts decoding from a single constant fully masked sentence and deterministically updates it at every iteration, the proposed SUNDAE is a probabilistic method which can leverage diversity of translations to boost quality (via rejection sampling) and can also generate high-fidelity texts unconditionally. Being probabilistic also entails differences in training: SUNDAE does not use mask-then-predict paradigm of SMART, rather starts with a random sentence and denoises it - more akin to multinomial diffusion methods. SMART only predicts masked tokens at generation time, so it needs to re-inject such tokens after the first step during training (otherwise there will be no masked tokens left which would create a significant train/test gap), while SUNDAE operates directly on noisy tokens and simply samples from the predicted transition distribution then feeds the samples back into the input.
>
> __Can you report the decoding speed to construct a complete comparison?__
> Addressed in the message ["Authors' response to all reviewers"](https://openreview.net/forum?id=T0GpzBQ1Fg6&noteId=WkxsAkcWC3V), please see above.
>
> __Does knowledge distillation bring benefits to your method and why?__
> Addressed in the message ["Authors' response to all reviewers"](https://openreview.net/forum?id=T0GpzBQ1Fg6&noteId=WkxsAkcWC3V), please see above. As for why, our method does model uncertainty to some degree (as unconditional evaluation shows), but probably not at the level of AR. This is likely why training with reduced uncertainty helps.
>
> __For the experiments of unconditional text generation, I suggest the authors provide quantitative evaluation scores to provide a clear comparison.__
> Addressed in the message ["Authors' response to all reviewers"](https://openreview.net/forum?id=T0GpzBQ1Fg6&noteId=WkxsAkcWC3V), please see above.
>
> We have added extra experimental results that were mentioned by the reviewer to the manuscript. If your concerns are alleviated by these extra results, we respectfully ask the reviewer whether they would change their rating.

---

### Official Review · Reviewer_D77g · 2021-11-02

**Correctness:** 3
**Technical Novelty And Significance:** 3
**Empirical Novelty And Significance:** 3
**Recommendation:** 6
**Confidence:** 4

**Main Review:**

#### **Strengths**:
1. The proposed method achieves new state-of-the-art performance in non-autoregressive machine translation tasks.
2. The paper is written well and easy to understand.

**My concerns concentrate on the experiments**.
1. Missing several important results. For example, a) efficiency analysis in machine translation tasks, we all know that deeper iterations improve the performance at the cost of the decoding efficiency, b) mere quantitative results in unconditional text generation tasks are helpless to understand whether it works well.
2. Some comparisons could not be fair—for example, the comparison with the Imputer model in Table 2. Although the performance improvement over the Imputer model is impressive, the proposed method (SUNDAE) jointly performs iterative decoding and parallel rerank techniques. However, the Imputer model does not use reranking techniques. In addition, we also notice that the Imputer model only needs fewer iterations.

#### **Questions**:
1. What do $x^{(N)}$ and $f_{\theta}^{(N)}$ in equation (2) mean?
2. Can the author use a picture to illustrate the target length prediction? It is also not clear to me why a different (to previous works) design is needed for length prediction.
3. Table 2 does not include the results of SUNDAE trained without knowledge distillation (KD). However, most baselines do not include performance without the KD. Why not report the SUNDAE's performances trained with the knowledge distillation for a comprehensive comparison?

**After reading the author's response, most of my concerns have been resolved in the revision. Therefore, I would like to increase my score by 6.**

**Summary Of The Paper:**

This paper proposes a new training technique for an iterative non-autoregressive (NAR) model and achieves significant improvements over the previous model. Inspired by the success of the diffusion model, the author formalizes the iterative NAR decoding process with a denoising autoencoder framework, then unrolls the denoising process during training. The proposed method (SUNDAE) achieves new state-of-art performances in machine translation by deep iteration decoding and parallel reranking. More experiments on unconditional text generation tasks further show its promising capability.

**Summary Of The Review:**

This paper proposes a new state-of-the-art performance in non-autoregressive translation without the help of an autoregressive teacher for training. However, some concerns about the experiment still need to be addressed to understand the contribution better.

---

> ### Author Response · Authors · 2021-11-20
> **Authors' response**
>
> We thank the reviewer for their work on the review.
>
> __Add efficiency analysis in machine translation tasks:__
> Addressed in the message ["Authors' response to all reviewers"](https://openreview.net/forum?id=T0GpzBQ1Fg6&noteId=WkxsAkcWC3V), please see above.
>
> __Mere qualitative results in unconditional text generation tasks don't help to understand whether it works well:__
> Addressed in the message ["Authors' response to all reviewers"](https://openreview.net/forum?id=T0GpzBQ1Fg6&noteId=WkxsAkcWC3V), please see above.
>
> __Some comparisons could not be fair—for example, the comparison with the Imputer model in Table 2. The Imputer model does not use reranking techniques. In addition, we also notice that the Imputer model only needs fewer iterations.__
> We have added (n = 1) to the table  to make it more explicit for the reader that Imputer doesn't use reranking. That said, we are explicit about using reranking and also report it for all methods in the table as (n = ...). Depending on the hardware, reranking n might be quite cheap compared to sequential steps T, because it could be done in parallel -- for example, reranking factor 16 can be easily batched on GPUs and TPUs if we need to serve a single translation quickly. As for the number of steps, in new distillation results (Table 1) SUNDAE can outperform Imputer with T=4 steps. We will make sure to clarify it in the text under what computational budget we outperform which methods.
>
> __What do x^N and f^N and in equation (2) mean?__
> x^N is the N-th dimension of x. f^N is the function which computes the probability distribution for this dimension. The whole formula is just to indicate that we use a factorized Markov transition distribution, just like multinomial diffusion methods do. We will make it more clear in the text.
>
> __Can the author use a picture to illustrate the target length prediction? It is also not clear to me why a different (to previous works) design is needed for length prediction.__
> We have now added such a picture (Figure 3). We used a different design, because it is easier to implement/maintain and less hand-crafted. Forcing the prediction to have a specific length is inelegant and brittle, better to allow the network itself to choose whether to use provided length or not. That said, we did try length-forcing early on in our experiments, which also worked.
>
> __Why not report the SUNDAE's performances trained with the knowledge distillation for a comprehensive comparison?__
> Addressed in the message ["Authors' response to all reviewers"](https://openreview.net/forum?id=T0GpzBQ1Fg6&noteId=WkxsAkcWC3V), please see above.
>
> We have added extra experimental results that were mentioned by the reviewer to the manuscript. If your concerns are alleviated by these extra results, we respectfully ask the reviewer whether they would change their rating.

---

### Official Review · Reviewer_Rdkb · 2021-11-07

**Correctness:** 3
**Technical Novelty And Significance:** 3
**Empirical Novelty And Significance:** 3
**Recommendation:** 8
**Confidence:** 3

**Details Of Ethics Concerns:**

Does not have any ethical concerns in my view.

**Main Review:**

Strengths:

The present approach (and approximations) are well motivated and show decent improvements over baselines on machine translation experiments.

Weaknesses/Questions:
1. While the improvements on "raw BLEU" are good, it is not clear why raw BLEU is interesting. If distilled models are able to provide good quality translation, why are methods to train from scratch required for conditional generation tasks like MT?
2. The motivation in the introduction does say that for unconditional generation methods, distillation is not possible where this method can be applied. To support this claim, some quantitative results should be provided (quality/diversity of the generated text) similar to what text GANs work report.
3. How sensitive is this approach to the heuristics and their related hyperparameter values (such as \rho, temperature etc)? Do they need to be tuned for every dataset/language pair for example?
4. The authors should also present an analysis of decoding speed of this method compared to baselines.

**Summary Of The Paper:**

This paper presents a training method of non-autoregressive text generation that is trained from scratch without distillation. The method is motivated as an unrolled denoising approach, which starts with a sequence of random tokens and transforms it in multiple stages to a valid sequence. To make training feasible, the proposed approach starts with a corrupted target sentence instead of a totally random sequence and does only two steps of unrolling. The training loss is an approximation of this process and is computed as a sum of loss on direct denoising and two steps of denoising. During inference, it starts with a random sequence and applies several steps of denoising to arrive at the desired sequence. Several heuristics are also applied to make generation feasible. The main result of the paper on non-autoregressive machine translation on which the authors show decent improvements over baselines on non-distilled settings however slightly underperforms AR distilled models. The authors also show examples for the case of unconditional generation and text in-painting

**Summary Of The Review:**

The paper shows good improvements on non-autoregressive translation without distilling from trained models, however, it is not clear why that is something to strive towards. The approach uses several heuristics and it is again not clear how sensitive this approach is to their values.

---

> ### Author Response · Authors · 2021-11-20
> **Authors' response**
>
> We thank the reviewer for their work on the review.
>
> __Not clear why raw BLEU is interesting:__
> Addressed in the message ["Authors' response to all reviewers"](https://openreview.net/forum?id=T0GpzBQ1Fg6&noteId=WkxsAkcWC3V), please see above.
>
> __Some quantitative results should be provided (quality/diversity of the generated text) similar to what text GANs work report:__
> Addressed in the message ["Authors' response to all reviewers"](https://openreview.net/forum?id=T0GpzBQ1Fg6&noteId=WkxsAkcWC3V), please see above.
>
> __How sensitive is this approach to the heuristics and their related hyperparameter values (such as \rho, temperature etc)? Do they need to be tuned for every dataset/language pair for example?__
> Usually the number of steps is determined by the computational budget (the more, the better results) and optimal temperature and \rho depend on it. We search the optimal parameters on the validation set for each language pair separately. In practice, the method is not very sensitive to hyperparameters though. For the temperature, please take a look at Figure 5b (added for the rebuttal): curves for different temperatures smoothly blend into each other as it is clear from the colors. All the good values are around temperature 0.2, because diversity is less important for translation. That said, very low temperatures kick off faster, but later get stuck. As for \rho, variance is slightly higher, but either 0.4 or 0.6 was usually the best.
>
> __Present an analysis of decoding speed of this method compared to baselines:__
> Addressed in the message ["Authors' response to all reviewers"](https://openreview.net/forum?id=T0GpzBQ1Fg6&noteId=WkxsAkcWC3V), please see above.
>
> We have added extra experimental results that were mentioned by the reviewer to the manuscript. If your concerns are alleviated by these extra results, we respectfully ask the reviewer whether they would change their rating.

---

> > ### Comment · Reviewer_Rdkb · 2021-11-29
> > **Thanks for all the changes!**
> >
> > I would like to thank the authors for making all the changes to their draft. Most of my questions/concerns have been addressed and I am increasing my score to 8.
> >
> > > distillation makes the training pipeline complicated and resource-consuming in practice because it requires sequentially decoding a pre-trained AR model on the entire training dataset (4.5M sentences). Moreover, the result of distillation is likely capped by the AR teacher quality - if we hope to overcome AR performance in the future, we need to train on raw data.
> >
> > Thank you for the clarification. I partially agree with the first statement however it is only a one-time complication.

---

### Author Response · Authors · 2021-11-20
**Authors' response to all reviewers**

We thank the reviewers for their work and their valuable comments. In the following, we will refer to reviews in the order they are shown on openreview: R1 = Rdkb, R2 = D77g, R3 = 7uVK, R4 = D7fL.

__Summary__:
We are happy that the reviewers find our method interesting and well-motivated (R1, R4), note significant improvement over the baselines (R1, R2, R4), comment positively on the quality of writing (R2, R3) and view our contributions as significant and new (R1, R2, R4).

As for the reviewers’ questions, we would like to highlight three key experiments we have now added to the manuscript:

__Report the SUNDAE's performances trained with the knowledge distillation for a comprehensive comparison (R1, R2, R3, R4)__:
Addressing this suggestion, we have added knowledge distillation experiments to our paper in Table 1, following the protocol previously applied by the Imputer paper [1]. To summarize the results, we now show significant improvements with respect to the baselines on EN -> DE among distillation-based non-AR results as well. That said, we believe non-distilled (raw) results originally presented in our paper are still valuable: distillation makes the training pipeline complicated and resource-consuming in practice because it requires sequentially decoding a pre-trained AR model on the entire training dataset (4.5M sentences). Moreover, the result of distillation is likely capped by the AR teacher quality - if we hope to overcome AR performance in the future, we need to train on raw data.

__Report the decoding speed to construct a complete comparison (R1, R2, R3)__:
Addressing this suggestion, we have added Table 3 with decoding speed, following the protocol in the Imputer paper [1]. The AR-relative speed-ups are in line with the Imputer paper for the same numbers of decoding steps.

__Quantitative results should be provided for unconditional generation (quality/diversity of the generated text) similar to what text GANs work report (R1, R2, R3, R4)__:
Addressing this suggestion, we have added in Figure 2 comparison to 5 text GAN methods using the established BLEU/self-BLEU methodology, well-described in ScratchGAN [2] and "Language GANs falling short" [3] papers. To summarize the results, our method outperforms 4 out of 5 GAN baselines, and ScratchGAN in the settings that value quality over diversity. The authors of ScratchGAN [2] were very kind to share their plotting code and details of the evaluation, so our numbers are directly comparable to those reported in their paper.

To address other questions, we will respond to each of the reviewers individually in comments to their reviews, and will further update the paper accordingly.

---

[1] Chitwan Saharia, William Chan, Saurabh Saxena, and Mohammad Norouzi. Non-Autoregressive Machine Translation with Latent Alignments. In Proceedings of the Conference on Empirical Methods in Natural Language Processing (EMNLP 2020). https://arxiv.org/abs/2004.07437

[2] Cyprien de Masson d’Autume, Mihaela Rosca, Jack Rae, and Shakir Mohamed. Training Language GANs from Scratch. In Proceedings of the Conference on Neural Information Processing Systems (NeurIPS 2019) https://arxiv.org/abs/1905.09922

[3] Massimo Caccia, Lucas Caccia, William Fedus, Hugo Larochelle, Joelle Pineau, and Laurent Charlin. Language GANs Falling Short. In Proceedings of the International Conference on Learning Representation (ICLR 2020) https://arxiv.org/abs/1811.02549

---

### Decision · Program_Chairs · 2022-01-20

**Decision:**

Accept (Poster)

**Comment:**

The paper introduces a simple technique to improve non-autoregressive generation by training the model to reconstruct model-perturbed inputs in addition to inputs perturbed by a fixed noise source.

Despite interest in the paper, we were worried about a number of aspects missing from section 3. During the rebuttal phase, however, the authors addressed most if not all comments and the section is now rather complete.

For its clarity, and the interesting results, we are recommending this version for acceptance.

A comment on presentation:

The paper attempts to establish a connection with variational diffusion models, but the connection does not seem strong enough at this point. In a variational diffusion approach, the forward view would not involve $f_\theta$, for example. Also, given that the distribution of $\mathbf x_t$ depends on $\theta$, the gradient estimator used in the paper is a heuristic, and I'd like to ask that the authors emphasise this clearly and early on in the draft.